# Parental Coping, Representations, and Interactions with Their Infants at High Risk of Cerebral Palsy

**DOI:** 10.3390/jcm12010277

**Published:** 2022-12-29

**Authors:** Silja Berg Kårstad, Åse Bjørseth, Johanna Lindstedt, Anne Synnøve Brenne, Helene Steihaug, Ann-Kristin Gunnes Elvrum

**Affiliations:** 1Regional Centre for Child and Youth Mental Health and Child Welfare (RKBU Central Norway), Department of Mental Health, Faculty of Medicine and Health Sciences, Norwegian University of Science and Technology, 7130 Trondheim, Norway; 2Child and Adolescent Mental Health Services, St. Olav’s Hospital, Trondheim University Hospital, 7130 Trondheim, Norway; 3Department of Psychology and Speech-Language Pathology, University of Turku, 20500 Turku, Finland; 4Department of Neuromedicine and Movement Science, Faculty of Medicine and Health Sciences, Norwegian University of Science and Technology, 7130 Trondheim, Norway; 5Department of Clinical and Molecular Medicine, Faculty of Medicine and Health Sciences, Norwegian University of Science and Technology, 7130 Trondheim, Norway; 6Clinical Services, St. Olav’s Hospital, Trondheim University Hospital, 7130 Trondheim, Norway

**Keywords:** CP, infant, stress, anxiety, depression, parents, fathers, parent–infant interaction, representations

## Abstract

The aim of this study is to describe parental coping, representations, and interactions during the time of inclusion in the Small Step early intervention program for infants at high risk of cerebral palsy (CP) in Norway (ClinicalTrials.gov: NCT03264339). Altogether, 11 infants (mean age 4.8 months, *SD*: 1.5) and their parents (mothers: *n =* 10, fathers:* n =* 9) were included. Parental coping was assessed using the Parenting Stress Index-Short Form (PSI-SF) and the Hospital Anxiety and Depression Scale (HADS). Parental representations and parent–infant interactions were assessed using the Working Model of the Child Interview (WMCI) and the Parent–Child Early Relational Assessment (PCERA). Parents’ PSI-SF and HADS scores were within normal range; however, 26.7% showed symptoms of stress, 52.6% showed symptoms of anxiety, and 31.6% showed symptoms of depression above the cut-off. WMCI results indicate that 73.7% of the parents had balanced representations. For PCERA, the subscale Dyadic Mutuality and Reciprocity was of concern, while two other subscales were in areas of strength and three subscales in some concern areas. There were no differences between mothers and fathers. Most of the parents had balanced representations, some had mental or stress symptoms and many were struggling with aspects of the parent–infant interaction. This knowledge could be useful when developing more family-centered interventions.

## 1. Introduction

Cerebral palsy (CP) results from a lesion or maldevelopment in the immature brain and is the most common severe motor disability in childhood [1]. The motor disorder is frequently accompanied by disturbances of cognition, communication, and epilepsy [2]. The birth prevalence varies from 1.4 to 2.5 per 1000 live births in high-income countries and is even higher in low- to middle-income countries [3,4,5,6,7]. About half of all infants with CP have identifiable risk factors in the newborn period, such as prematurity, low birthweight for gestational age, genetic abnormalities, or encephalopathy [8]. New guidelines recommend diagnosing high risk of CP at 4 to 6 months so that interventions can start as early as possible [8,9]. Recent research is focusing on habilitation services that provide more accurate knowledge about the psychological needs of parents and their infants at high risk of CP in order to develop more family-centered habilitative interventions [10,11].

Becoming a parent can be a stressful experience that demands great responsibility for the newborn child and can cause concerns about their development and health [12,13,14,15,16,17]. However, when risk of brain damage occurs during pregnancy, labor, or shortly after birth, parents are substantially more prone to experiencing high levels of stress [18,19,20], which might lead to the development of mental health problems such as depression and anxiety [21,22,23]. Parents of infants at high risk of disease are often hospitalized with their newborn, and it may be traumatic to witness their infant experience various medical procedures and assessments [24]. Worries for potential sequala or diagnosis may put an additional strain on parents [25]. Studies of older children with CP have found that mothers experience more stress than fathers [26], while the experience of stress among mothers and father of infants diagnosed with a high risk of CP is largely unknown.

Parents of children and adolescents with CP are found to have an increased risk for mental health problems. This has been proposed in a systematic review [23], which indicates that symptoms of depression and anxiety are more prevalent in parents of children with CP compared to healthy controls. The results from this review suggest that the severity of a child’s condition and the time required to care for the child are risk factors for developing mental health problems. Another review reveals that parental coping ranges from parents who do not perceive their child’s disabilities as stressful, to parents who report negative stress and describe their lives as challenging [26]. Thus far, few studies have described parental coping at the time when their infant is diagnosed with a high risk of CP, yet most of these studies include mothers. One recent study from Sweden describes that almost one-third of mothers with an infant at high risk of CP scored above the cut-off value for symptoms of anxiety and depression [27]. This percentage exceeds what has been reported in population-based studies without any known risk factors, showing a prevalence of 15% for postnatal anxiety symptoms for mothers [13] and 7% for fathers [28], as well as 11.9% for depression during the perinatal period for mothers [29] and 3–5% for fathers [30,31]. There is a need to investigate further the anxiety and depression rates in mothers and fathers of infants with a high risk of CP to gain knowledge regarding potential risks for mental health problems.

Parents’ relationship with their infants may also be at risk when parents experience stress and worries concerning their infants’ health [32,33,34,35]. Typically, the development of a parent–infant relationship begins before the infant is born through the parents’ mental representations of themselves, combined with their thoughts and feelings about the unborn child [36,37,38]. Parental representations assessed with the Working Model of the Child Interview (WMCI) are classified into the global categories: balanced, disengaged, or distorted [39]. It has been found that in clinical populations of infants and toddlers at risk, or with a diagnosis, most parents’ representations were disengaged (34.2%) and distorted (43.6%) [32]. To our knowledge, no studies have assessed parents’ mental representations at the time their infant was given the diagnosis at high risk of CP. Some studies have found more balanced representations in mothers of full-term infants compared to mothers of pre-term infants using the WMCI [35,40], while other studies indicated no differences [33,41]. Furthermore, a few studies have found a relationship between non-balanced representations in mothers and higher levels of depression both in clinical and non-clinical samples [33,34]. Getting to know more about parents’ representations of their infant at high risk of CP may help professionals set up suitable support and interventions adjusted to the needs of the families [42].

Despite the knowledge that fathers spend much time with their infants, few studies have focused on the father–infant relationship. A recent longitudinal study of fathers with typical developing infants showed that higher levels of sensitivity, and lower levels of withdrawal behaviors were more often observed in fathers with balanced compared to unbalanced prenatal representations [38]. In another longitudinal study of fathers of typical developing infants, they found that early attachment representations of the infant predicted the quality of future father–infant interaction [43]. Similar findings have been found with mothers showing positive relationship between balanced representations and better quality of infant-mother interaction [44,45]. Since there are few studies including both genders, the present study will describe the representations of both mothers and fathers of infants at high risk of CP and investigate if there are any differences.

The quality of parent–infant interactions contribute to an infant’s cognitive, emotional, and social development [46,47]. Previous studies have investigated parent–infant interactions when the infant is at risk of different conditions [45,48,49,50,51,52,53], and for parents with, or at risk of, mental health problems [54,55,56,57]. In a recent review of the parent–infant interaction of infants at risk of CP compared to healthy populations, it was found that infants at risk were generally less active and showed fewer facial expressions. Furthermore, mothers were more intrusive, and parent–infant dyads were described as less synchronized, with fewer sensitive responses [48]. However, the studies included in that systematic review did not use the new, recommended guidelines for setting a diagnosis with a high risk of CP [8]. Rather, prematurity was used as the main inclusion criteria to indicate high risk in several studies; only one study included fathers, and they found no differences between the interaction qualities of mothers and fathers [58]. Thus, there is an urgent need for studies that investigate the early dyadic interaction between parents and their infant who is diagnosed with a high risk of CP. The present study aims to (1) describe parental coping, parental representations, and parent–infant interaction during inclusion in the Small Step early intervention program for infants at high risk of CP (i.e., when the infant is between 4 and 6 months old), and (2) assess if there are differences between mothers and fathers in coping, representations, and the quality of parent-infant interaction.

## 2. Materials and Methods

### 2.1. Design

This study is part of the Small Step early intervention study performed at St. Olavs hospital, Trondheim, Norway from September 2017 to July 2020 (ClinicalTrials.gov: NCT03264339). The study was performed in collaboration with the researchers who developed the Small Step early intervention program at the Karolinska Institute, Sweden [27,59]. In the Small Step study, a single subject research design was used, with each participant serving as his/her own control through multiple testing at baseline and during intervention and withdrawal periods [60]. In the current study, data collected through the baseline period were applied. The Small Step study was approved by the Regional Ethical Committee (REC) for Medical Research in Mid-Norway (2016/1366).

### 2.2. Participants

Eligible participants were families with an infant diagnosed with CP, or at high risk of CP, at the regular clinical follow-up at three months corrected age for infants with known complications before, during or shortly after birth. The guidelines for setting the diagnosis with a high risk of CP were used, i.e., assessment of general movements (GMs), neonatal magnetic resonance imaging (MRI), and neurological assessment with Hammersmith Infant Neurological Examination (HINE) [8]. In addition, motor development was assessed using the Alberta Infant Motor Scale (AIMS) [61]

### 2.3. Procedurals

During the baseline period, parental coping, parental representations, and parent-infant interactions were assessed once for each parent. In addition, the infants were tested at three time-points with various motor tests that are outside the scope of the current study. The testing took place either in the family’s home or at the hospital. 

*Parental coping* was measured with the Parenting Stress Index-Short Form third edition (PSI-SF) [62]. The PSI-SF is a 36-item, self-report measure of parenting stress where parents rate items on a 5-point scale. The PSI-SF includes a Total Stress scale and three subscales: Parental Distress, Parent–Child Dysfunctional Interaction and Difficult Child. In the present study, we used the Total Stress scale. The Total Stress scale ranges from 36 to 180 and is seen as an indicator of a parent’s overall experience of parenting stress. The 90th percentile of the PSI-SF score represents a “clinically significant” level of parenting stress and can be used as an indicator that counseling or other support is required. The PSI-SF demonstrates high internal consistency, test–retest reliability, and validity [63,64,65].

In addition, we used the Hospital Anxiety and Depression Scale (HADS) to investigate parental coping [66]. The HADS consists of 14 questions, seven measure symptoms of anxiety and seven measure symptoms of depression. Each question has four answer categories ranging from zero to three, where category three indicates the highest level of the symptom. The HADS is divided into a scale for anxiety (HADS-A) and a scale for depression (HADS-D), with scores ranging from 0 to 21. Scores between 0 and 7 are within the normal range, while scores between 8 and 10 indicate mild symptoms, scores between 11 and 14 indicate moderate symptoms, and scores between 15 and 21 indicate severe symptoms. The HADS has been shown to have a good factor structure, discriminant validity and internal consistency [67,68].

*Parental representations* of their infant were assessed with the Working Model of the Child Interview (WMCI) [39]. The WMCI is a semi-structured interview where caregivers are asked about their subjective experiences and perceptions of their child, parenting and their relationship with the child. The caregiver’s narratives are classified into six qualitative scales (i.e., Richness of Perceptions, Openness to Change, Intensity of Involvement, Coherence, Caregiving Sensitivity, and Acceptance). High scores in the qualitative scales indicate positive parental narrative qualities, except for the scale of Intensity of Involvement, where a score of 3 is the most optimal. The WMCI also includes two content scales (i.e., Infant Difficulty and Fear for Safety), where high scores represent negative parental narrative content. In addition, the caregiver’s affective tone of the representations is coded, identifying how much joy, pride, anger, disappointment, anxiety, guilt, indifference, or other emotions were expressed throughout the interview. Parents’ representations were classified into three main categories (balanced, disengaged or distorted). The two latter categories can be classified as non-balanced representations. The WMCI has good psychometric properties, and the reliability of the clinical scales is found to be satisfactory in a Norwegian sample of infants [32,69]. The WMCI interviews lasted approximately 30–90 min and were videotaped and scored by certified coders not involved in the intervention study. The main coder (A.S.B) scored all the 19 interviews using a 5-point Likert scale and another certified coder (Å.B) scored six interviews (30%). They agreed upon the main categories in 4 of the 6 interviews corresponding to an interrater agreement of 0.67. The two interviews that were coded differently were discussed and consensus was made by the two coders on both categorical and scale levels. Thus, the main coder’s scorings were used for 17 of the interviews and the consensus scores for two interviews.

*Parent–infant interactions* were assessed during five minutes of videotaped free play using the Parent–Child Early Relational Assessment (PCERA) [70]. The PCERA is widely used in the Nordic countries as an observation method that measures the quality of affect and behavior in parent–infant interactions and it is shown to have acceptable psychometric properties [38,50,51,54,71]. The videos were recorded either in the family’s home or at the hospital. The parents were filmed with their infants on separate occasions. The parents received the following instruction: “Play with your infant as you normally do. You can use the toys if you like or play without the toys”. The PCERA free play situations were rated according to the manual [70] by two trained coders. The complete PCERA consists of 65 independent items. In the present study, 60 PC-ERA items were rated since some of the items are not ratable for infants under 6–9 months. The main coder (A.-K.G.E.) rated all video recordings, and 20% of the videos were double rated by another certified coder (J.L.). To retain interrater agreement, drift sessions between coders were held throughout the assessment process. All PCERA items were rated on a five-point Likert scale. The coders considered the frequency, duration, and intensity of the behaviors when rating each item. After rating all the videos, the coders decided that the item “mirroring” had to be removed, since five parents used their native language in the play situation. Thus, it was difficult to score “mirroring”, i.e., parent’s attunements with their child’s emotional state, including consideration of parents labeling of their infants’ internal feeling state. Before the interrater agreement calculation, all items were recoded into a three-point scale describing areas of concern (scores 1 and 2), areas of some concern (score 3) and areas of strength (scores 4 and 5), as conducted in previous studies [38,50,54]. The interrater agreement was calculated by the mean percentile of the raters’ overall agreement. The interrater agreement between the two coders was 0.80, which is considered acceptable [72]. Before analyzing the data, 52 items were combined into 6 subscales using the five-point scale according to the “4 Month feeding factors” described in the PCERA manual [70]: (1) Parent Positive Affective Involvement, Sensitivity, and Responsiveness; (2) Parent Negative Affect and Behavior; (3) Infant Positive Affect, Communicative and Social Skills; (4) Infant Dysregulation and Irritability; (5) Dyadic Mutuality and Reciprocity; and (6) Dyadic Tension. High PCERA scores indicate positive affect or/and behavior; therefore, high scores on subscales 2, 4, and 6 indicate a lack of negative affect and/or behavior. In this study, Cronbach’s *α* coefficients for calculating the internal consistency of the six PCERA subscales ranged between excellent and acceptable [73]: 0.95 (subscale 1), 0.89 (subscale 2), 0.85 (subscale 3), 0.90 (subscale 4), 0.83 (subscale 5), and 0.78 (subscale 6).

### 2.4. Statistical Analysis

We used the IBM SPSS statistics 27 program to analyze the data [74]. Descriptive statistics with means, standard deviations, confidence intervals and percentages were used to present the data. The variables were normally distributed according to Q–Q plots and we used a paired-sample *t* test to compare means between mothers’ and fathers’ coping scores, qualitative and content WMCI scores, and the PCERA subscales. An alpha level of 0.05 was used to determine statistical significance.

## 3. Results

### 3.1. Sample Characteristics

Altogether, 19 parents (10 mothers and 9 fathers) of 11 infants from ten families consented to participate in the Small Step early intervention study. Two more families were invited, however one family declined due to long travels, and the other family could not be included because of the involvement from the child protective service. All 11 infants had a clinical history indicating a risk for CP. Four of the infants were born premature. Among these, three were twins and one was born extremely premature because of placental abruption. One of the infants born at term had microcephaly, two had difficult births causing asphyxia, and four had various complications within the first ten days after birth. At the regular clinical hospital follow-up at three months corrected age, the infants had the following high-risk factors for CP: absent (*n* = 6) or sporadic (*n* = 5) fidgety GMs, suboptimal HINE scores < 57 (*n* = 10), abnormal MRI (*n* = 11), and delayed motor gross motor skills, as indicated by AIMS scores at or below the fifth percentile (*n* = 6) or between the fifth and tenth percentile (*n* = 5). 

The baseline testing was performed during three timepoints between two and six weeks after the time of diagnosis, except for one infant where summer holidays delayed the baseline testing (age at first baseline: 5.7 months) and another infant where hospital stays prolonged the baseline period till 10 weeks. See Table 1 describing parent and infant characteristics. About 2/3 of the parents had university degrees, equally distributed between mothers (*n* = 6) and fathers (*n* = 6). Five of the parents were not native Norwegians; therefore, three of the WMCI interviews were conducted in English, and two with an interpreter. 

### 3.2. Parental Coping

The PSI-SF was completed by 17 of the 19 parents, and two forms were excluded because of missing scores. For four forms the score for one item was missing and imputation was done according to the manual [62]. For two forms there were missing scores for three and five items, respectively, and we decided to exclude these forms. The results for the remaining 15 parents indicate that the parents’ mean stress scores were within normal range; however, ¼ showed stress above normal range in the clinically area. There was no significant difference between mothers’ and fathers’ mean stress scores. Parents’ mean anxiety and depression scores measured with the HADS were within normal range, although as shown in Table 2, almost half of the parents (52.6%) scored above cut-off for anxiety (from score 8) and 31.6% showed depression symptoms. Mothers’ mean scores for anxiety and depression were higher than fathers’ mean scores, but these differences were not significant. 

### 3.3. Parental Representations

The results from the WMCI interviews indicate that, at the categorical level, 73.7% (*n* = 14) of parents’ representations were balanced, 21.1% (*n* = 4) were disengaged, and 5.2% (*n* = 1) were distorted. The mean scores for the eight WMCI scales are shown in Table 3.

As a group, all means for the qualitative and content scales shown in Table 3 are within the non-concerned range. Counting the frequency of the individual parents’ mean scores on the eight scales reveals that 0–15.8% were in the concerned range on the qualitative scales and 5.3–10.6% on the content scales. Furthermore, the following affective contents of the representations during the interview were most common: joy (*mean* = 3.0, *SD* = 1.1), pride (*mean* = 3.3, *SD* = 1.3) and sadness/sorrow (*mean* = 4.1, *SD* = 2.2). We found no significant differences between mothers’ and fathers’ mean scores on the qualitative or content scales with independent *t*-tests. 

### 3.4. Parent–Infant Interaction

The mean scores for the PCERA subscales are presented in Table 4. The high mean scores on parental subscales 2 (Negative Affect and Behavior) and infant subscale 4 (Dysregulation and Irritability) indicate strength areas in parent–infant interactions, suggesting low levels of negative affect both in parents and infants. For the parent subscale 1 (Positive Affective Involvement, Sensitivity and Responsiveness), infant subscale 3 (Positive Affect, Communicative and Social Skills) and parent–infant subscale 6 (Dyadic Tension), mean scores indicate areas of some concern, while the mean score for parent–infant subscale 5 (Dyadic Mutuality and Reciprocity) indicates an area of concern. 

Figure 1 shows the six subscales with a percentage distribution for the three PCERA categories. As seen in the figure, most of the parents were in the “some concern” area on subscales 1, 3, 5 and 6, and the concern area was highest in subscales 1 and 5.

To describe the parent–infant interactions in more detail, we report in Table 5 the items that showed mean scores above four in the strengths area (16 of 52 items) and items that had mean scores below three (6 of 52 items), indicating items of concern. High item scores indicate positive affect or/and behavior or a lack of negative affect and/or behavior. 

We found no significant differences between mothers’ and fathers’ mean parent–infant interaction subscale scores.

## 4. Discussion

Our results indicate that parents’ mean stress, anxiety and depression symptoms were within normal range. However, some parents reported symptoms of stress (26.7%), anxiety (52.6%, including both borderline and abnormal scores) and depression (31.6%, including both borderline and abnormal scores) above normal level. Parents’ representations of their infant were primarily balanced (73.7%), and there were low levels of negative affect in both parents and infants. However, there were some concerns regarding parents’ affective involvement, sensitivity, and responsiveness towards their infants, as well as the infants’ communicative and social competence. This may have affected the parent–infant dyad, causing some tension between the infant and the parent, and decreasing dyadic mutuality and reciprocity. We found no significant differences between mothers and fathers in coping, representations, and the quality of parent-infant interaction.

There was a large variation in reported stress levels and symptoms of anxiety and depression among the parents in our study. For most of the parents, feelings of stress, anxiety, or depression were within normal levels. Additionally, we found no differences between mothers’ and fathers’ levels of stress, anxiety, and depression symptoms. Knowing that all the included infants had a history of complications before, during, or shortly after birth, and that they had recently received the diagnosis at high risk of CP, may indicate that there are factors beyond birth-related trauma and receiving a diagnosis that affected coping in this group of parents. This is in accordance with a review that demonstrated great differences in how parents deal with having a child with CP [26]. Some parents may have protective factors such as being in a stable relationship and having a supportive family that makes them more resilient to traumatic experiences and more able to cope with stress and trauma [26,75]. Furthermore, the relatively high educational level among the parents in the current study might be a protective factor [26]. Nonetheless, about one quarter of the included parents had stress symptoms in clinical areas, and symptoms of anxiety and depression were similar or higher than in comparable populations [27] and much higher than reported in populations without any known risk factors [13,28,29,30,31]. This indicates that it is important to assess the levels of stress, anxiety, and depression in parents of infants at risk of CP and to promote potential protective factors to increase parental coping [75].

According to parental representations, assessed with the WMCI, the percentage of balanced representations (73.7%) in this study was higher than for parents of prematurely born infants (20–55%) [35,40,50] and for parents of low- to moderate-risk infants in Norway (58.3%) [69]. One possible explanation for these differences may be a lower number of parents in the present study compared to previous studies [33,35,40]. Another explanation might be that nearly half of the infants in the current study (*n* = 4) were born at full term after non-complicated pregnancies. It is possible that the parents of these infants had already developed balanced representations during pregnancy, and research has shown that these representations are often quite stable [37]. Additionally, there were no differences between the mothers’ and fathers’ representations. This may indicate that mothers and fathers develop similar patterns of thoughts and feelings towards their relationship with their infant at high risk of CP, however this finding needs to be replicated with studies with larger number of participants

In this study, the parents displayed little, or no displeasure, annoyance, frustration, or anger towards their infant, and the infants were easily soothed by their parents. This is considered an interactive strength and may suggest that having an infant at high risk of CP does not increase the risk of physical abuse [76]. Nonetheless, the decreased mutuality and reciprocity of the interaction was of concern, with low mean scores for turn taking, mutual enthusiasm, joyfulness, and enjoyment. This could be due to several infants having decreased communicative skills, with reduced or delayed responses to social initiatives made by the parents. Similar results have been found in other studies, indicating that reduced communicative abilities affect the parent–infant dyad [34,48]. Motor impairments and the reduced quality of the exploratory play could place additional limitations on playfulness and enjoyment in the interaction. Our results also indicate some concerns when it comes to the parents’ sensitivity and responsiveness towards their infant. Previous studies reported that mothers of infants at risk of CP were less sensitive and demonstrated less smiling compared to mothers of healthy controls [48,49,77]. Few studies have investigated the dyad between fathers and infants, but one study from Finland investigated this for fathers of typically developing infants during their first months of life [38]. In this Finnish study, similar patterns to those in our study for areas of strengths and concerns were identified, except with less dyadic tension in the parent–infant dyad for fathers of typically developing infants. In our study, we did not find any differences between the scores for mothers and fathers. Thus, it seems that the concerns identified in our study may be related to having an infant at high risk of CP and not related to the fact that we included both mothers and fathers. 

Overall, our results indicate that there may be both protective and risk factors in our sample of parents of infants at high risk of CP. Most of the parents had balanced representations, and stress, anxiety, and depression symptoms were within normal range. However, some parents experienced mental health problems or stress above normal level, and many struggled with aspects of parent–infant interactions. Thus, our results underscore the need for assessments, preventive strategies, and interventions, not only focusing on the infants, but also targeting parents’ stress levels, mental health, and interactions with their infant.

One of the strengths of this study was the use of diverse assessments, such as self-report questionaries, interviews, and observations, to illuminate parental coping, representations, and interactions with their infants at high risk of CP. This provides a broader understanding of the psychological challenges parents may encounter when having an infant at high risk of CP and how this may affect parent–infant relationships. Secondly, our sample included fathers, which are seldom studied in research on postnatal mental health, and even more lacking in studies of infants at risk [26,31]. Recent research shows that some fathers have mental health problems regarding the parenting of healthy babies [28,31,78]; therefore, it is important to understand more about fathers of infants at high risk of CP. The health system in Norway continues to mainly assess mothers’ mental health on a regular basis, so we need increased attention to the fact that fathers also may struggle. Finally, our study included infants at very high risk of CP according to recent diagnostic guidelines [8]. Most previous studies included premature infants, with unknown additional risks for CP.

Our study also has some limitations. We only recorded parent–infant interactions during play activities using the PCERA. It would have been preferrable to include recordings of an additional situation, for example feeding or diaper changing. However, it was necessary to reduce the number of assessments to avoid fatigue for both the infants and the parents. Furthermore, in some of the parent-infant situations the other parent was present, in addition to the one who was filming. It is possible that this affected the behavior of the parent being filmed, however there was no interaction between the parents, and the other parent was in another section of the room. Additionally, we did not double code all the WMCI videotapes because double coding all instruments was time-consuming and expensive. This could possibly have strengthened the presentation of the level of interrater reliability. Finally, the cross-sectional design used in this study precludes any causal implications, and because of the low number of participants, we cannot generalize our findings. Thus, our findings need to be further investigated in a larger sample, including both mothers and fathers of infants at high risk of CP. In addition, we need more longitudinal studies following infants and parents over an extended period to describe the trajectories of their mental health and quality of interactions.

## Figures and Tables

**Figure 1 jcm-12-00277-f001:**
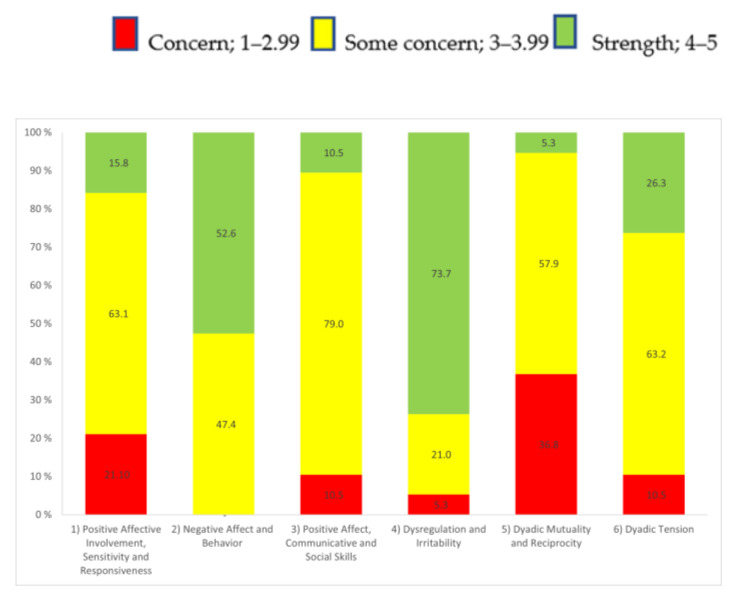
The six subscales with percentages in the three Parent–Child Early Relational Assessment categories: concern, some concern, and strengths.

**Table 1 jcm-12-00277-t001:** Sample characteristics parents (*n* = 19) and infants (*n* = 11).

Characteristics Parents		Total *n* = 19	Mothers *n* = 10	Fathers *n* = 9
Language (*n*)				
	Norwegian	10	7	7
	Other *	5	3	2
Mean age in years (min-max)		34 (25–57) ^2^	32 (25–42) ^1^	36 (28–57) ^1^
Highest degree (*n*)				
	Upper secondary school	3	3	0
	Vocational training	4	2	2
	Bachelor’s degree	7	4	3
	Master’s or doctoral degree	5	2	3
**Characteristics infants**		**Total *n* = 11**		
Gender: Female/Male (*n*)		4/7		
Gestational age				
	Mean gestational age: weeks (range; ± SD)	35.6 (24.6–41.4; ± 5.3)		
	Term/preterm (*n*)	7/4		
Risk factors for cerebral palsy (*n*)				
	GMs: absent/sporadic fidgety movements	6/5		
	HINE scores: suboptimal < 57/normal	10/1		
	MRI: abnormal	11		
	AIMS: ≤ 5^th^ percentile/5^th^ - 10^th^ percentile	6/5		
Additional impairments (*n*)				
	Epilepsy	2		
	Cortical visual impairment	2		
	Hearing impairment	2		
	Hydrocephalus, shunt	2		
	Nasogastric intubation	2		
	Bronchopulmonary dysplasia	1		
Corrected age at baseline				
	Mean age baseline 1: months (range; ± SD)	4.2 (3.5–6.7; ± 0.9)		
	Mean age baseline 2: months (range; ± SD)	4.8 (4.0–7.2; ± 0.8)		
	Mean age baseline 3: months (range; ± SD)	5.3 (4.5–7.7; ± 0.9)		
Family (*n*)				
	Living with both parents: Yes/No	9/2		
	Number of siblings: 0/1/2/3	1/8/1/1		

Note: *n* = number, * = English as second language, min = minimum, max = maximum, ^2^ = age missing for *n* = 2; ^1^ = age missing for *n* = 1; SD = Standard Deviation, GMs = General Movements, HINE = Hammersmith Infant Neurological Examination, MRI = Magnetic Resonance Imaging, AIMS = Alberta Infant Motor Scale.

**Table 2 jcm-12-00277-t002:** Percent and mean scores of the Parenting Stress Index-Short Form third edition (PSI-SF) and the Hospital Anxiety and Depression Scale (HADS) for the group and for the mothers and fathers.

Variable	Percentages (*n*)	Mean (*SD*)	95% CI
*Parenting stress*			
PSI-SF total ^1^	100 (15)	77.4 (26.5)	61.7–93.5
Fathers		78.0 (27.6)	56.9–99.1
Mothers		76.5 (27.5)	47.5–105.5
Low level ^2^	26.7 (4)		
Normal level ^2^	46.6 (7)		
High level ^2^	0		
Clinically ^2^	26.7 (4)		
*Anxiety (HADS-A) ^3^*			
Total	100 (19)	7.5 (4.9)	4.0–9.7
Fathers		6.2 (4.6)	2.7–9.8
Mothers		8.6 (5.1)	4.9–12.3
Normal	47.4 (9)		
Borderline	15.8 (3)		
Abnormal	36.8 (7)		
*Depression (HADS-D) ^3^*			
Total	100 (19)	5.3 (4.2)	2.4–7.5
Fathers		3.8 (3.3)	1.3–6.3
Mothers		6.7 (4.5)	3.5–9.9
Normal	68.4 (13)		
Borderline	15.8 (3)		
Abnormal	15.8 (3)		

Note: *SD* = Standard deviation. CI = Confidence interval. *n* = number. ^1^ PSI-SF = Parenting Stress Index-Short Form. ^2^ PSI-SF Total scores: 36–55 = low stress, 56–85 normal stress, 86–90 = high stress and above 90 = clinically significant stress level. ^3^ Anxiety and Depression total scores: 0–7 = Normal, 8–10 = Borderline, and 11–21 = Abnormal. HADS = Hospital Anxiety and Depression Scale, D = depression and A = anxiety.

**Table 3 jcm-12-00277-t003:** Mean scores of the Qualitative and Content Scales of the Working Model of the Child Interview (WMCI) measuring parents’ representations of their infants for the group and for the mothers and fathers.

WMCI	*n*	Mean (*SD*)	95% CI
Qualitative Scales			
Richness of Perceptions	19	3.7 (1.0)	3.2–4.1
Fathers	8	3.9 (0.4)	2.9–4.8
Mothers	11	3.6 (0.3)	2.9–4.2
Openness to Change	19	3.7 (1.0)	3.3–4.2
Fathers	8	4.0 (0.3)	3.2–4.8
Mothers	11	3.6 (0.3)	2.9–4.2
Intensity of Involvement	19	3.6 (1.1)	3.1–4.1
Fathers	8	3.9 (0.4)	2.9–4.8
Mothers	11	3.5 (0.3)	2.8–4.1
Coherence	19	4.0 (0.9)	3.5–4.4
Fathers	8	4.1 (0.3)	3.4–4.8
Mothers	11	3.8 (0.3)	3.2–4.4
Caregiving Sensitivity	19	3.9 (0.9)	3.5–4.3
Fathers	8	4.1 (0.2)	3.6–4.7
Mothers	11	3.7 (0.3)	3.1–4.4
Acceptance	19	3.8 (1.0)	3.4–4.3
Fathers	8	3.9 (0.4)	3.1–4.7
Mothers	11	3.8 (0.3)	3.1–4.5
Content Scales			
Infant Difficulty	19	2.4 (1.1)	1.8–2.9
Father	8	2.5 (0.3)	1.7–3.3
Mother	11	2.3 (0.4)	1.4–3.1
Fear for the Infant’s Safety	19	2.7 (0.9)	2.3–3.1
Father	8	2.5 (0.3)	1.9–3.1
Mother	11	2.8 (0.3)	2.2–3.5

Note: *SD* = Standard deviation. CI = Confidence interval. *n* = number. High scores in the qualitative scales indicate positive parental narrative qualities, except for the scale of Intensity of Involvement, where a score of 3 is the most optimal. The two content scales (i.e., Infant Difficulty and Fear for Safety) high scores represent negative parental narrative content.

**Table 4 jcm-12-00277-t004:** The Parent–Child Early Relational Assessment subscales mean scores measuring the parent–infant interactions for the group and for mothers and fathers.

Subscales	*n*	Mean (*SD*)	95% CI
Parental Positive Affective Involvement, Sensitivity and Responsiveness	19	3.5 (0.5)	3.2–3.7
Fathers	8	3.3 (0.2)	2.9–3.8
Mothers	11	3.6 (0.1)	3.3–3.9
Parental Negative Affect and Behavior	19	4.0 (0.4)	3.7–4.1
Fathers	8	3.8 (0.2)	3.5–4.2
Mothers	11	4.1 (0.1)	3.8–4.3
Infant Positive Affect, Communicative and Social Skills	19	3.3 (0.5)	2.9–3.5
Fathers	8	3.4 (0.2)	3.0–3.8
Mothers	11	3.2 (0.1)	2.9–3.5
Infant Dysregulation and Irritability	19	4.1 (0.4)	3.8–4.2
Fathers	8	4.1 (0.1)	3.9–4.3
Mothers	11	4.0 (0.1)	3.7–4.3
Dyadic Mutuality and Reciprocity	19	2.9 (0.6)	2.5–3.2
Fathers	8	3.0 (0.3)	2.4–3.6
Mothers	11	2.9 (0.1)	2.5–3.2
Dyadic Tension	19	3.5 (0.5)	3.2–3.7
Fathers	8	3.5 (0.2)	3.1–4.0
Mothers	11	3.5 (0.2)	3.2–3.9

Note: *SD* = Standard deviation. CI = Confidence interval. *n* = number. Scores 1 and 2 describes areas of concern, score 3 means areas of some concern and scores 4 and 5 are areas of strength.

**Table 5 jcm-12-00277-t005:** The Parent–Child Early Relational Assessment items mean scores in the strength and concern area.

Items		*n*	Mean (*SD*)	95% CI
*Parental items strength area*				
	Annoyed, Angry, Hostile Tone of Voice	19	4.9 (0.3)	4.7–5.1
	Warm, Kind Tone of Voice	19	4.2 (0.7)	3.8–4.5
	Expressed Negative Affect	19	4.4 (0.6)	4.1–4.7
	Irritable/Frustrated/Angry Mood	19	4.9 (0.2)	4.8–5.1
	Depressed Mood	19	4.2 (0.8)	3.8–4.6
	Displeasure, Disapproval, Criticism	19	4.6 (0.5)	4.3–4.8
	Negative Physical Contact	19	4.4 (0.7)	4.0–4.7
	Amount of Visual Contact with Child	19	4.4 (0.5)	4.1–4.6
	Responsivity to Child’s Negative or Unresponsive behavior	19	4.3 (0.6)	4.0–4.6
*Infant items strength area*				
	Apathetic, Withdrawn, Depressed Mood	19	4.0 (0.7)	3.7–4.3
	Irritable/Frustrated/Angry Mood	19	4.2 (0.5)	4.0–4.5
	Emotion Lability	19	4.9 (0.2)	4.8–5.1
	Robustness	19	4.3 (0.6)	4.0–4.6
	Consolability/Soothability	19	4.6 (0.7)	4.1–5.1
*Dyad items strength area*				
	Frustrated, Angry, Hostile	19	4.5 (0.5)	4.2–4.7
	Tension, Anxiety	19	4.0 (0.7)	3.6–4.4
*Parent items concern area*				
	Amount of Verbalization	19	2.8 (0.8)	2.4–3.6
*Infant items concern area*				
	Social Behavior of Infant-Initiates	19	2.9 (0.9)	2.5–3.4
	Quality of Exploratory Play	19	2.8 (1.0)	2.3–3.3
	Communicative Competence	19	2.7 (0.6)	2.4–3.0
*Dyadic items concern area*				
	Mutual Enthusiasm, Joyfulness, Enjoyment, Dyadic “Joie de Vivre”	19	2.8 (0.7)	2.5–3.1
	Reciprocity	19	2.7 (0.7)	2.4–3.0

Note: *SD* = Standard deviation. CI = Confidence interval. *n* = number.

## Data Availability

The data presented in this study are available on request from the corresponding author. The data are not publicly available due to ethical restrictions.

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
