# Peer review of "Parental Coping, Representations, and Interactions with Their Infants at High Risk of Cerebral Palsy"

_jcm, 2022, doi:10.3390/jcm12010277_

Round 1

Reviewer 1 Report

SUMMARY

The study aims to (1) describe parental coping, parental representations, and parent–infant interaction of infants at high risk of CP, and (2) assess if there are differences between mothers and fathers in coping, representations, and the quality of parent-infant interaction.

The fist aim is achieved by presenting descriptive statistics and the second one by performing paired-sample t test. The authors conclude that, as a group, mental health symptoms of parents were within normal range, but some parents presented stress, anxiety and depression above normal level. Parents’ representations of their infant were primarily balanced. However, there were some concerns about parents’ affective involvement, sensitivity, and responsiveness, as well as the infants’ communicative and social competence. This affected the parent–infant dyad, causing some tension between the infant and the parent, and decreasing dyadic mutuality and reciprocity.

GENERAL COMMENTS

The manuscript is clear, relevant for the field and presented in a well-structured manner. The study is very well written and methos are presented transparently and are reproducible. The research is relatively novel as includes children diagnosed with CP, or at high risk of CP with mean age <6mo and considers parent representations. The topic is clinically relevant as allows better understanding of the nature and origin of social competence among children with CP and potential targets for intervention. Conclusions are mostly consistent with the evidence but I would recommend some edits to ensure that the way they are worded cannot be misinterpreted (see below comments about likes 320-321).

The main strength of this study is the comprehensive assessment including not only parental coping measures but also representations and parent infant interactions. It is also noteworthy the inclusion of fathers and the inclusion of complementary qualitative information, which can be quite informative in studies of this nature.

An important design limitation, however, is the small sample size (n=11 infants, n=19 parents). Results should therefore be taken with caution and more details about sample characteristics and their heterogeneity/homogeneity provided. Moreover, no control group is included, and normative data is used instead, which should be considered when interpreting the results.

SPECIFIC COMMENTS

Introduction

Line 41: I would recommend to also referring to a more recent and global study of CP prevalence: McIntyre S, Goldsmith S, Webb A, Ehlinger V, Hollung SJ, McConnell K, et al. Global prevalence of cerebral palsy: A systematic analysis. Developmental Medicine & Child Neurology. 2022 Aug 11;64(1494–506).

Line 49: “Becoming a parent can be a stressful experience that …. causes concerns about their development and health [11-16].” Please, consider changing the term “causes” by “can cause” as this would be more accurate, particularly considering citations 11-16.

Lines 53-54: Could please provide a reference to support the following statement: “may be traumatic to witness their infant experience various medical procedures and assessments”?

Line 74: Could you explain what exactly needs to be investigated further and how your study aligns with this? The topic is relevant and, of course, would benefit from further research in general, but could you please be more specific in your rationale?

Lines 75-89: I would recommend mentioning which role may parental representations play on the interactional quality and other relevant outcomes in light of the available literature. The interest on this specific issue is not directly justified and some relevant literature could be introduced here (consider, for example Korja et al 2010, Sokolowski 2007, among others).

Materials and methods

Did you undergo a power analysis before performing the study?

Participants: As a suggestion, the sample characteristics and other details related with this (lines 129 to 146) may be better placed in the results section. These aspects were not planned methodological aspects but is the description of the sample once applying the preestablished criteria.

Information about the homogeneity between participants in terms of clinical characteristics is of great interest, especially given the small sample size. Could you please add the following information?

- How many weeks passed between the time of the diagnosis and baseline testing in the two participants for whom the baseline period was prolonged?

- Age range

- Sex

              - GA (term/preterm)

- Were there complications during pregnancy? (Some of this information is in discussion, but should be presented here too)

- How may were diagnosed with CP, and how many at high risk of CP?

I strongly suggest presenting this (and other information that is currently in the manuscript) in a table so relevant data can be easily checked by the readers.

Could you please describe if the used standardized assessments control for socioeconomic or education level?

Line 130: Could you make clear if the 11 infants were from different families?

Lines 207-208: Could you please explain what do you mean by language issues? This could be informative for clinicians aiming to use the PCERA and other researchers willing to replicate the work.

Line 169: Please check the following minor gramma error “HADS are…”

Results

Line 234: Did you register why scores were missing? If so and it is related with the characteristics of the assessment, I would recommend providing additional details.

Discussion

Results related with Aim 2 should also be presented within this first paragraph of the discussion. 

Could you please discuss with your results what you presented in lines 70-74 of the introduction?

It should be discussed how the lack of a control group can affect the results.

Line 315: Could you please check whether 26.8% should be 26.7%?

Line 316: You should make clear that the percentages that you are providing include both borderline and abnormal scores joined.

Lines 320-321: While the proposed causality between variables is interesting and likely true, this should not be presented as a result but as a hypothesis.

While the small sample size limits the possibility of undergoing analysis considering the level of education of participants, I would encourage briefly mentioning how this could have influence the reported outcomes.

Tables

Tables are not as easy to interpret as they could be. I would undergo some minor edits towards making them clearer. For example, consider the following:

-          Table 1 states insight the table that PSI-SF is being presented but this is not done for the HADS. Although this is clear in the caption and table footer that the table includes both PSI-SF and HADS information, this can be a bit confusing.

-          Could you include in the 2 and 3 the ranges for interpreting the WMCI and PCERA respectively? So, the sable can be self-informative for readers who are not familiar with these scales.

-          Table 3: Include the name of the subscales (instead of numbers) within the table as done with the other tables.  

Abstract

Minor comment: I recommend not using the symbol n to refer to the number of scales, while it is understandable this could be confusing for the readers. Consider rephrasing the sentence of line 28 by “while 2 others were in areas of strength…” 

Author Response

"please see attachment"

Reviewer 2 Report

It was with great pleasure that I reviewed the manuscript entitled "Parental Coping, Representations, and Interactions with their infants at high risk of cerebral palsy".

I offer some suggestions for the authors.

Introduction

1) Can the authors provide some examples of "mental health problems" that may occur in parents of children at high risk for CP (line 52)?

2) Although the purpose of this study is interesting, the "need for further investigation" in this area is not clear to the reader. The introduction describes in detail previous findings on this topic. I encourage the authors to further highlight the lack of knowledge justifying this study and its originality.

Materials and Methods

1) Participants: Lines 129-146 are the results and should therefore be moved to the Results section.

2) The authors should justify the double scoring of only 30% (WMCI) and 20% (PCERA) of the videotaped tests. For the WMCI, an inter-rater agreement could complete the results and reassure the reader.

Results

1) It might be interesting to have the specific scores of mothers and fathers in the tables. Since comparison is a secondary objective, it is frustrating to have to look for these partial results in the text.

2) The WMCI results in Table 2 show parent difficulties in the categories of "Infant Difficulty" and "Fear of Infant's Safety" which, without error on my part, are not included in the text or commented on in the discussion. Yet they seem to be an important part of understanding the profile of parents of infants at high risk for CP.

3) I encourage the authors to find a way to provide a visual representation of the results described in lines 298-312. These results appear to be an original description of the parents' profile. 

4) With this amount of assessment, exploring different facets of the parent profile, I think the authors should use advanced statistical analysis to explore correlations and/or relationships between variables to further push the relevance and originality of the results (for example: the authors could explore WMCI and/or PCERA factors that influence anxiety or depression). Results will be preliminary given the small number of participants but may provide new insights for future research.

Discussion

While the results remain in this configuration (without further exploration), the discussion is well linked to them and is appropriately structured. The clinical implications are limited but cannot be taken further at this stage.
